# A centrifugation-based physicochemical characterization method for the interaction between proteins and nanoparticles

Ahmet Bekdemir[1] & Francesco Stellacci[1,2]

Nanomedicine requires in-depth knowledge of nanoparticle–protein interactions. These interactions are studied with methods limited to large or fluorescently labelled nanoparticles as they rely on scattering or fluorescence-correlation signals. Here, we have developed a method based on analytical ultracentrifugation (AUC) as an absorbance-based, label-free tool to determine dissociation constants ($K_D$), stoichiometry ($N_{max}$), and Hill coefficient ($n$), for the association of bovine serum albumin (BSA) with gold nanoparticles. Absorption at 520 nm in AUC renders the measurements insensitive to unbound and aggregated proteins. Measurements remain accurate and do not become more challenging for small (sub-10 nm) nanoparticles. In AUC, frictional ratio analysis allows for the qualitative assessment of the shape of the analyte. Data suggests that small-nanoparticles/protein complexes significantly deviate from a spherical shape even at maximum coverage. We believe that this method could become one of the established approaches for the characterization of the interaction of (small) nanoparticles with proteins.

[1] Institute of Materials, Ecole Polytechnique Fédérale de Lausanne (EPFL), CH-1015 Lausanne, Switzerland. [2] Interfaculty Institute of Bioengineering, EPFL, CH-1015 Lausanne, Switzerland. Correspondence and requests for materials should be addressed to F.S. (email: francesco.stellacci@epfl.ch).

Colloidal nanoparticles (NPs) are of broad interest in terms of their interactions with biomolecules. In particular, when in contact with biological fluids, they are eventually surrounded by plasma proteins that form a 'protein corona'[1,2]. These NP—protein complexes can alter the intended function of NPs[3,4]. It is clear that there is a need to measure and understand with a certain precision the physical-chemistry principles that underline the interaction of proteins with NPs. Initially, this is best achieved with simple and well characterized NPs and model proteins.

The estimation of equilibrium constants as well as stoichiometry of protein-NP complexes have been a centre of interest for more than a decade in NP-protein research[5–9]. For example, fluorescence based techniques have been used to monitor the emission of protein solutions in presence of metal NPs. The signal is typically the autofluorescence of tryptophan that is quenched with increasing NP concentration. The decrease in fluorescence is described by a Langmuir isotherm to estimate the dissociation constant ($K_D$) and Hill coefficient ($n$)[10]. These methods, however, have some shortcomings. For example, fluorescence quenching is NP's size dependent[11], NP concentration determination, in most cases, is not straightforward[12] and large protein aggregates can cause spurious signals.

Techniques based on the diffusion coefficient of species, such as dynamic light scattering (DLS)[13,14] or fluorescence correlation spectroscopy (FCS)[8], are also widely used. In these methods, optical signals (scattering or fluorescence) trace the Brownian motion of NP-protein complexes in solution. The translational diffusion coefficient ($D$) is then estimated and converted into the hydrodynamic diameter of the object through the Stokes-Einstein equation. A gradual increase in hydrodynamic diameter ($d_H$) of NPs upon protein adsorption implies the formation of a protein corona[6]. These methods are fast and available to many users. On the other hand, particularly for scattering based methods, when the NP size is comparable to the protein size, the signal of the complete solution is dominated by the free proteins. Although this could be overcome with FCS by selectively labelling the NPs, this entails additional fluorescent molecules on the NPs that could alter the interaction with proteins.

Analytical ultracentrifugation (AUC) is a fractionation based technique that has been extensively used to characterize proteins and ultra small colloidal systems for almost a century[15,16]. It allows for the determination of the size, density, shape and heterogeneity of bio/nano-materials in solution with only one experiment without any standards. AUC employs various optical measurement systems such as absorbance, fluorescence or interference. It is based not only on diffusion but also on sedimentation of the species. Using these two analytically decoupled parameters together, AUC is able to estimate hydrodynamic parameters with excellent accuracy, even for non-spherical species[15].

Here we present absorbance-based AUC as a robust tool for the physicochemical characterization of the interactions of NPs with a protein in solution. We provide examples where all the common thermodynamic interactions parameters for the association of gold NPs (AuNPs) with bovine serum albumin (BSA) are determined. This method successfully estimates $K_D$, the maximum number of proteins per NP ($N_{max}$) and $n$ for the non-specific binding of BSA to AuNPs with a core diameter bigger than 10 nm, or as small as 2.2 nm. To the best of our knowledge, the latter data are reported for the first time. We find that an optical-absorption based technique such as AUC has several advantages: the insensitivity towards large (protein) aggregates present in solution, the limited dependence on particle size, and the possibility of working in many different media.

In AUC, a homogenous solution is placed in a cell and once equilibrated, is subject to rotation at a constant angular velocity $\omega$. When centrifugal force is applied to NPs or proteins in solution, they sediment through the cell and create a concentration gradient as well as a diffusion flux[17]. All concentration distributions of the analyte $c(r, t)$ with respect to time ($t$) and to distance from rotor centre ($r$) are overlaid and their shape and time-evolution can be modelled by the Lamm equation (1), where sedimentation ($s$) and diffusion ($D$) coefficients can be decoupled. Both $s$ and $D$ can be solved with different numerical analysis methods such as the moving hat function method[18] or the adaptive space-time finite element method[19]. Recent advancements in computational power together with improvements of calculation methods considerably decreased the analysis time.

$$\frac{\partial c}{\partial t} = D\left(\frac{\partial^2 c}{\partial r^2} + \frac{1}{r}\frac{\partial c}{\partial r}\right) - \omega^2 s\left(r\frac{\partial c}{\partial r} + 2c\right) \qquad (1)$$

AUC experimental pathways can be divided into two categories: sedimentation velocity (SV) and sedimentation equilibrium (SE). In SV experiments, analytes are centrifuged under high speeds and completely removed from solution by sedimentation. The resulting analysis provides size and shape information about species inside the solution through $s$ and $D$ parameterization. Recently, this method was also used to characterize size, density and molecular weight of organic soluble gold nanoclusters[20]. SE experiments employ lower speeds and attain the 'equilibrium' of sedimentation and back-diffusion of the analyte in solution. These experiments are more effective, although limited to extremely monodisperse systems, for determination of molecular weight ($M_w$) of proteins as well as investigation of protein–protein interactions[21,22]. Data analysis software available such as Ultrascan[23] or SEDFIT[17,24] provide user-friendly platforms to assess the quality of data and pre-defined models for homogeneous/heterogeneous protein–protein interactions with different stoichiometry. Most of the models, however, require well-defined molecular weight ($M_w$) and extinction coefficients ($\varepsilon$) of the interacting species. Most NPs, on the other hand, have diverse $M_w$ and $\varepsilon$. Therefore, determination of the stoichiometry of protein–NP complexes can not be accurately determined with established methods. Here, we developed a Langmuir adsorption model based on sedimentation coefficients of NP-protein complexes estimated from individual SV experiments. We also show that the simultaneous extraction of $s$ and $D$ information from SV experiments allows a qualitative assessment of shape changes during the absorption process for various size AuNPs.

## Results

**Theoretical foundations**. In a conventional SV experiment, a particle in a solution is subjected to three different forces that come into equilibrium and reach terminal velocity in a short period of time. Centrifugal force is directed towards the bottom of the cell and depends on the rotor acceleration ($\omega^2 r$) while the buoyant force and the frictional force act in the opposite direction. These forces are summed up in the Svedberg equation and combined with Stokes-Einstein relation, eventually providing the Stokes-equivalent spherical diameters[16]:

$$r_H = \frac{1}{2}\sqrt{\frac{18\eta s}{(\rho_p - \rho_s)}} \qquad (2)$$

where $r_H$ is the hydrodynamic radius of an analyte, $\eta$ is the solvent viscosity, $s$ is the sedimentation coefficient, $\rho_p$ and $\rho_s$ are the densities of the analyte and solvent, respectively. The density

of a NP–protein complex varies depending on the ratio of each component. Assuming the formation of homogeneous aggregates at equilibrium, we can express the density of the aggregate as shown in equation (3). The density and volume information of both the isolated NPs and proteins can be derived from individual SV experiments.

$$\rho_{cx} = \frac{\rho_{Np}V_{Np} + N_{avg}\rho_{P}V_{P}}{V_{Np} + N_{avg}V_{P}} \quad (3)$$

$N_{avg}$ is the average number of protein per NP, $V_{Np}$ and $V_{P}$ are the volumes of NP and protein, respectively and $\rho_{cx}$, $\rho_{Np}$, $\rho_{P}$ are the densities of NP–protein complex, NP and protein alone, respectively. When equations (2) and (3) are combined, the average sedimentation coefficient of the NP—protein complex can be expressed in terms of $N_{avg}$:

$$s_{cx}(N_{avg}) = \frac{2}{9\eta}\sqrt[3]{\frac{9}{16\pi^2}}\frac{\left(\rho_{Np} - \rho_{s}\right)V_{Np} + N_{avg}(\rho_{P} - \rho_{s})V_{P}}{\left(V_{Np} + N_{avg}V_{P}\right)^{1/3}} \quad (4)$$

$N_{avg}$ can be expressed as a function of protein concentration according to the following equation[8]:

$$N_{avg} = N_{max}\frac{1}{1 + (K_{D}/[P])^{n}} \quad (5)$$

where $N_{max}$ is maximum number of protein per particle, $K_{D}$ is dissociation constant of interaction and $n$ is the Hill coefficient, and $[P]$ is the concentration of the protein studied. Therefore, using equation (5) with (4) provides a complete model that is to be used for fitting the experimental data obtained from SV-AUC experiments for AuNPs together with BSA as a model protein:

$$s_{cx}([BSA]) = \frac{2}{9\eta}\sqrt[3]{\frac{9}{16\pi^2}}\frac{\left(\rho_{Np} - \rho_{s}\right)V_{Np} + N_{max}\frac{1}{1 + (K_{D}/[BSA])^{n}}(\rho_{P} - \rho_{s})V_{P}}{\left(V_{Np} + N_{max}\frac{1}{1 + (K_{D}/[BSA])^{n}}V_{P}\right)^{1/3}} \quad (6)$$

During the fitting of an adsorption isotherm, the parameters $K_{D}$, $N_{max}$ and $n$, in equation (6), are varied while all the other parameters $\rho_{NP}$, $V_{NP}$, $\rho_{P}$ and $V_{P}$ are obtained from separate SV-AUC experiments and kept constant. The adsorption isotherm derived here employs the Hill equation. The latter requires the adsorption to be reversible within the timeframe of the equilibration employed in the experiment (here 16 h). An analysis of the reversibility of the adsorption in our system is presented in the Results section of this paper.

**NP synthesis and characterization.** Three types of gold NPs were synthesized in this work: citrate coated (citrate-AuNPs), 11-mercaptoundecanoic acid-coated (MUA-AuNPs) and 11-mercaptoundecane sulfonate coated (MUS–AuNPs). Two separate synthetic methods were applied to obtain different sized

MUS–AuNPs which are hereafter referred to as MUS(m)–AuNPs for medium sized and MUS(s)–AuNPs for small sized NPs.

Synthesis of water-soluble small AuNPs often produces broad size distribution[25]. Since AUC is sensitive to even small variations in size and density, post-treatment to narrow the size distribution was necessary so as to interpret and construct the method more reliably. Before starting the measurements with proteins, we employed density gradient ultracentrifugation method (DGU) to fractionate NPs with respect to size and density (Methods section). The quality of the fractions was examined with SV-AUC $c(s)$ and the most abundant fraction for each synthesis was used. The characterization of fractions used hereafter is summarized in Table 1.

**Determination of interaction parameters.** First, we started with MUA–AuNPs as these particles are neither large nor small and are suitable candidates for comparison with previously reported values[8]. NPs solutions at a constant concentration were incubated with several concentrations of BSA and SV experiments were conducted for each mixture (refer to Supplementary Table 1 for the NPs concentrations and Supplementary Note 1 for the approximate calculation of NPs' molecular weight). The $c(s,D)$ distributions for each mixture was calculated in SEDFIT and plotted together to construct the absorption isotherm as shown in Fig. 1a. The experimental data was fitted to equation (6) and the interaction parameters for MUA–AuNPs with BSA were estimated (Fig. 1b). In the literature the interaction of HSA with FePt NPs had been studied[8]. In the Table 2 we show that NPs with similar hydrodynamic diameter (9.4 nm for MUA-AuNPs and 11.2 nm for FePt NPs) and similar ligand type (carboxylic acid terminated) lead to comparable protein adsorption behaviour in terms of $K_{D}$ and $n$. The subtle difference in $N_{max}$ values might be due to the variation in size as lower $N_{max}$ value corresponds to smaller AuNPs. However, the possibility of different ligand packing densities for NPs of different metals should also be taken into account to reach the final conclusion.

The second type of NPs used were 14 nm citrate coated AuNPs. The dissociation constant, $K_{D}$ calculated with our AUC method for the interaction of citrate-AuNPs and BSA interaction was found to be $13.6 \pm 3.5 \times 10^{-6}$ M (Fig. 2a). This value is close to the previously reported value (association constant, $K_{A} = 0.05 \pm 0.01 \times 10^{6}$ M$^{-1}$) obtained using quartz crystal microbalance (QCM)[26].

To work with NPs with different surface functional groups, we investigated MUS(m)–AuNPs (slightly smaller NPs than MUA–AuNPs) ($d_{H} = 7.6$ nm) (Fig. 2b). The residuals plot shows that excellent fitting quality with feasible interaction parameters are achieved regardless of the surface functionality (Fig. 2c). Although this work is based on BSA as a model protein, other proteins could also be investigated once their density and hydrodynamic volume are known. We show here that the method is also valid when studying NPs interaction with HSA, see Table 3, (Supplementary Fig. 1 for adsorption isotherm). In addition, Table 3 summarizes the variation in Hill coefficients

**Table 1 | Summary of NP properties.**

|  | $d_{core}$ (nm) | $d_{H}$ (nm) | PDI (%) | Ligand type | Ligand length (nm) |
|---|---|---|---|---|---|
| Citrate–AuNPs | 12.6 | 13.6 | 12 | Citric acid | 0.5 |
| MUA–AuNPs | 6.1 | 9.4 | 10 | − COOH | 1.5 |
| MUS(m)–AuNPs | 4.4 | 7.0 | 11 | − SO$_2$OH | 1.6 |
| MUS(s)–AuNPs | 2.2 | 5.6 | 16 | − SO$_2$OH | 1.6 |

AuNP, association of gold nanoparticles; MUA, 11-mercaptoundecanoic acid; MUS, 11-mercaptoundecane sulfonate; PDI, polydispersity index; 3D, three dimensional.
Polydispersity Index (PDI) was estimated via the s.d. of the Stokes radius distribution from $c(s)$ distribution in SEDFIT software. Hydrodynamic diameters and densities of AuNPs are calculated according to a previously reported method[20]. Ligand lengths were calculated with ACDLabs 3D chemical drawing software assuming fully extended conformation of the ligands. Ligand length information is only provided for comparison and were not used in any of the calculation in this work.

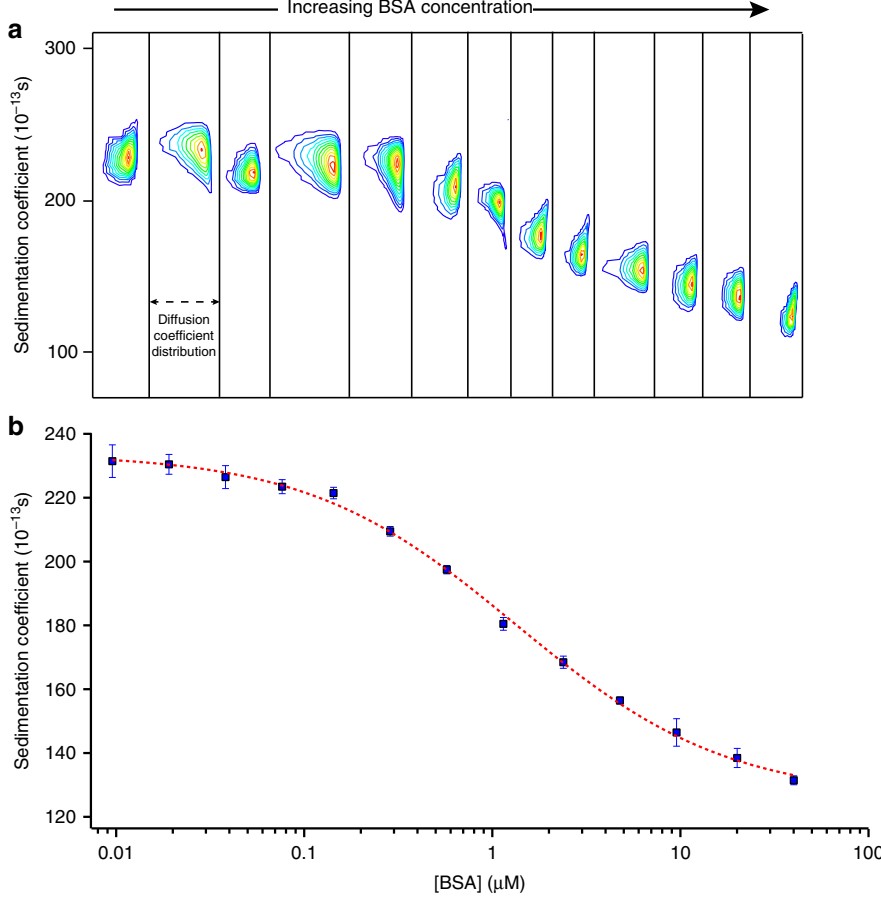

**Figure 1 | SV–AUC experiments for MUA–AuNPs with BSA.** (**a**) Concentration contour graphs of 2D-$c(s,D)$ analysis with the $y$ axis representing the sedimentation ($s$) values. (**b**) A plot showing the average s values (blue squares) of each individual graphs shown in (**a**) plotted against the protein concentration. The red dashed line is the fit function obtained using equation (6). Error bars represent the s.d. of three independent experiments from the same batch of NPs.

**Table 2 | Interaction parameters for MUA-AuNPs with BSA by AUC and carboxylic acid coated FePt NPs with HSA by fluorescence correlation spectroscopy (FCS)[8].**

| | $d_H$ (nm) | $K_D$ ($10^{-6}$ M) | $N_{max}$ | $n$ |
|---|---|---|---|---|
| AUC (MUA-AuNPs) | 9.4 | 5.4 ± 0.9 | 18 ± 2 | 0.8 ± 0.1 |
| FCS (FePt NPs)[8] | 11.2 | 5.1 ± 1.3 | 22 ± 4 | 0.7 ± 0.1 |

AUC, analytical ultracentrifugation; AuNP, association of gold nanoparticles; BSA, bovine serum albumin; HSA, human serum albumin; MUA, 11-mercaptoundecanoic acid.

depending on the particle type. Citrate and MUA coated AuNPs show an anti-cooperative ($<1$) effect while sulfonate functionalized particles show a cooperative effect. A discussion of the meaning of these observations is beyond the scope of this paper, however, we have to point out that a cooperative Hill coefficient has already been observed in some nanomaterials interaction with proteins[27,28].

As discussed above, the validity of our approach relies on the adsorption being reversible within the equilibration time of the experiment. To test for this key assumption, we performed AUC on an MUS(m)–AuNPs–BSA mixture equilibrated for 16 h. We then diluted the mixture 10 fold and left to equilibrate for another 16 h. The AUC of this diluted mixture showed a shift towards higher $s$ values in the $c(s)$ distribution of MUS(m)–AuNPs–BSA complexes indicating the spontaneous desorption of some proteins from initial protein-NP complex (see Supplementary

Fig. 2 and Fig. 3). This validates the use of the Hill formalism for the study of this problem. We should point out on the other hand, that not every type of protein–NP interactions will show this (or any) degree of reversibility. It is reported that some larger polystyrene particles show irreversible hard corona formation with transferrin[29].

AUC also allows investigation of very small colloidal particles. To prove this point, we synthesized and fractionated MUS(s)–AuNPs selecting a fraction with 2.2 nm core diameter. The $c(s)$ analysis of MUS(s)–AuNPs and BSA complexes were modelled through the fit function with an excellent fit quality (Fig. 3). To the best of our knowledge, investigation of the thermodynamic parameters of interaction with proteins with such small particles is presented here for the first time.

The limitation when investigating small sized NPs with AUC depends on the sedimentation coefficient. The proteins under investigation should have sufficiently distinct $s$ from the $s$ of NPs, otherwise the changes in sedimentation coefficient as a function of protein concentration would be too small. In most cases, this does not pose any problem because most proteins have low $s$ compared to metal NPs due to their low densities (1.2–1.3 g cm$^{-3}$).

In summary, this approach could be used for most types of NPs as long as they are suitable for AUC optics systems–absorbance, fluorescence, interference, etc. It is hard to provide exact ranges of binding constants and NPs sizes where this technique can be applied, as it depends on the difference in

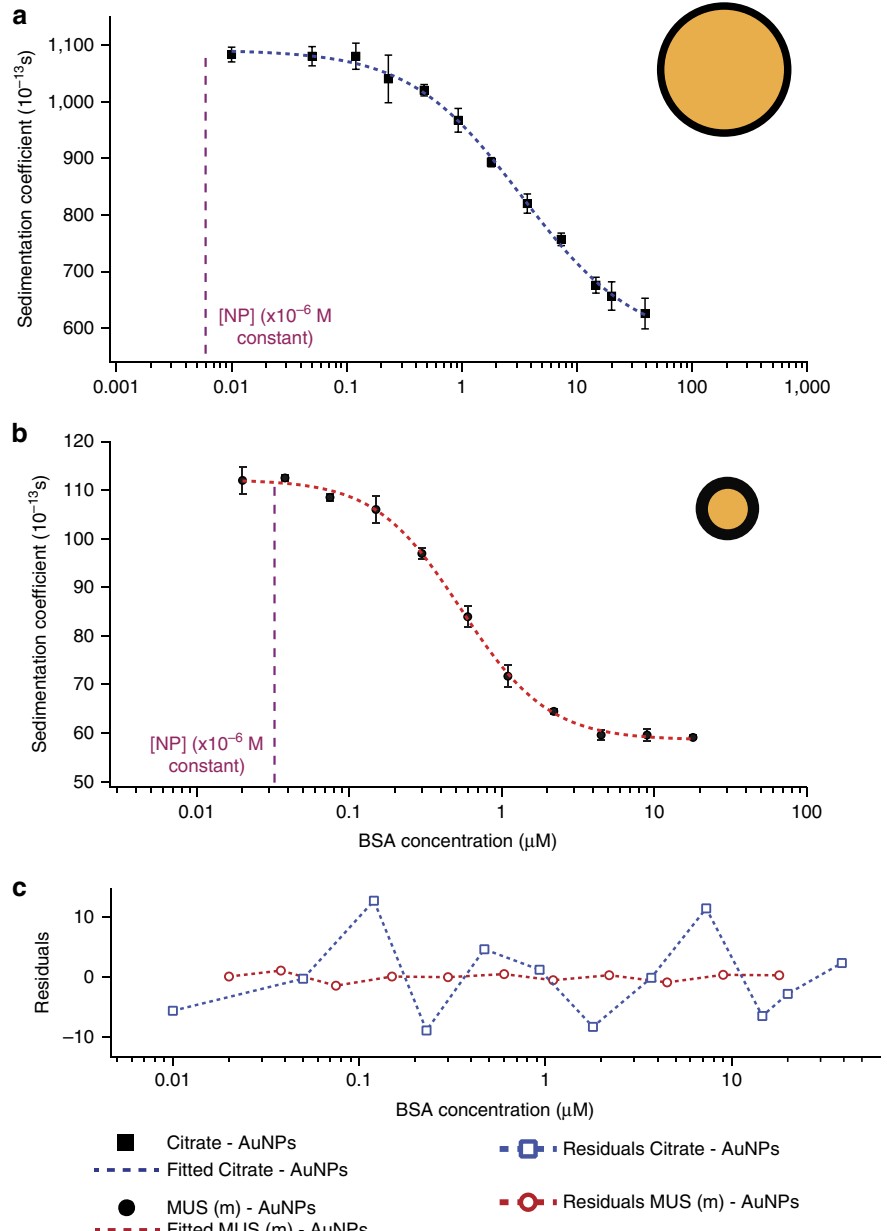

**Figure 2 | SV-AUC binding isotherms for Citrate-AuNPs and MUS(m)-AuNPs with BSA.** Plots of average $s$ versus BSA concentration for (**a**) citrate–AuNPs ($d_H = 13.6$ nm) and (**b**) MUS(m)–AuNPs ($d_H = 7.6$ nm). Dashed lines are fits obtained using equation (6). (**c**) Residuals of the fit functions are plotted against BSA concentration. Dashed magenta line indicates the nanoparticle concentration kept constant during experiments. Error bars represent the standard deviation of three independent measurements from the same batch of NPs.

**Table 3 | Interaction parameters obtained from AUC for various types of AuNPs and proteins.**

|  | $d_{core}$ (nm) | $d_H$ (nm) | $\rho$ (g cm$^{-3}$) | $s$ (S) | $D$ (m$^2$ s$^{-1}$) | $K_D$ ($10^{-6}$ M) | $N_{max}$ | $n$ |
|---|---|---|---|---|---|---|---|---|
| Citrate AuNPs–BSA | 12.6 | 13.6 | 17.7 | 1,080 | $3.38 \times 10^{-11}$ | $13.6 \pm 3.5$ | $36 \pm 5$ | $0.9 \pm 0.1$ |
| MUA AuNPs–BSA | 6.1 | 9.4 | 5.9 | 234 | $4.13 \times 10^{-11}$ | $5.4 \pm 0.9$ | $18 \pm 2$ | $0.8 \pm 0.1$ |
| MUS(s) AuNPs–BSA | 2.2 | 5.6 | 2.4 | 22 | $6.16 \times 10^{-11}$ | $9.2 \pm 0.9$ | $2 \pm 0$ | $1.6 \pm 0.2$ |
| MUS(m) AuNPs–BSA | 4.4 | 7.0 | 5.5 | 119 | $5.78 \times 10^{-11}$ | $1.1 \pm 0.1$ | $10 \pm 1$ | $1.3 \pm 0.1$ |
| MUS(m)–AuNPs–HSA | 4.4 | 7.0 | 5.5 | 119 | $5.78 \times 10^{-11}$ | $0.57 \pm 0.1$ | $5 \pm 1$ | $1.6 \pm 0.3$ |

AUC, analytical ultracentrifugation; AuNP, association of gold nanoparticles; BSA, bovine serum albumin; HSA, human serum albumin; MUA, 11-mercaptoundecanoic acid; MUS, 11-mercaptoundecane sulfonate.

density between particles and proteins. Any size range, then, should be specified with respect to the core material of the NP. As an indication for gold NPs, we can state that the lower limit of a

$K_D$ measurable would be $\sim 0.01\,\mu$M. This is because at this value the binding of the proteins will happen at approximately the NPs concentration, invalidating one key assumption in Hill equation

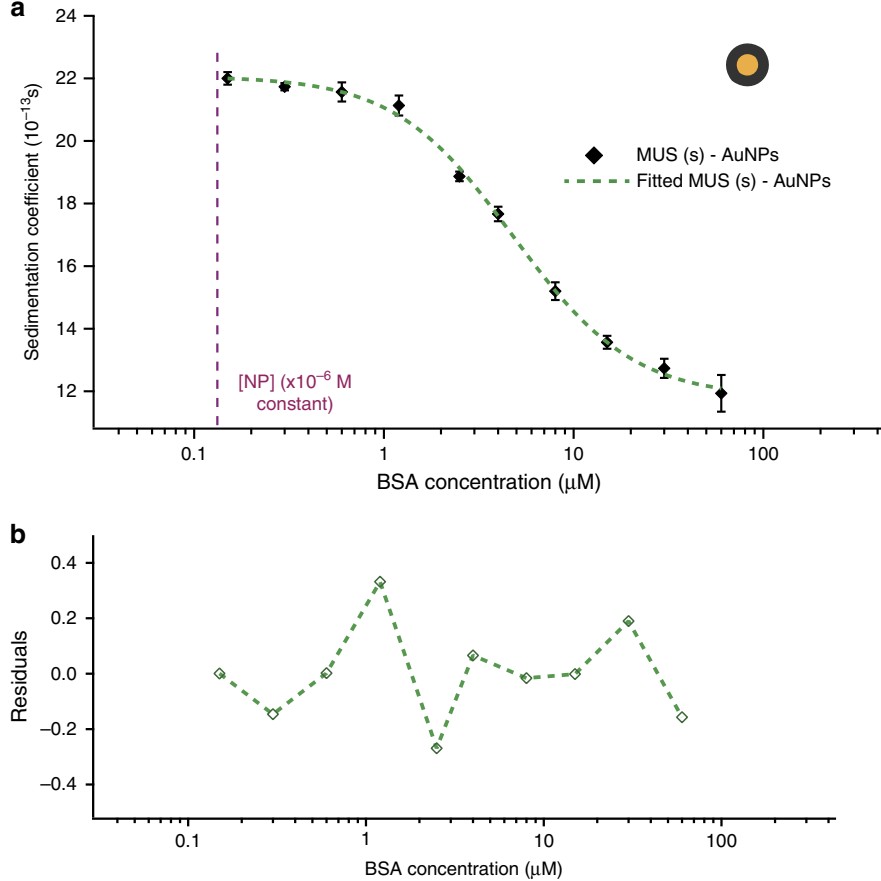

**Figure 3 | SV-AUC binding isotherm for very small AuNPs.** (**a**) Plot of average $s$ versus BSA concentration for MUS(s)–AuNPs ($d_H = 5.6$ nm). Error bars represent the standard deviation of three independent measurements from the same batch of NPs. (**b**) Residuals plot of the fit function of MUS(s)–AuNPs–BSA isotherm. Dashed magenta line indicates the nanoparticle concentration kept constant during experiments.

that the concentration of the proteins is in large excess. Given that the NP concentration is determined by their absorbance (the optical density of the solution should be in the range of 0.2 to 1.2 for a reliable optical measurement), as particle concentration varies, so does the lower limit for $K_D$. The upper limit of $K_D$ is mostly independent of NPs but depends on the nature of the protein. One should keep the viscosity of the protein solution in mind, as viscosity could substantially affect the process of sedimentation of the solutes. Based on this consideration, we arbitrarily estimate an upper limit of protein concentration to be $\sim 10$ mM. Thus, $K_D$ value could be around 1 mM at the highest. As discussed above, the limitation of the method in terms of NP size is directly related to colloidal stability of the NPs. In our experience, gold NPs from 2 to 25 nm core size are easily measurable in AUC. We believe the above discussion can give the reader an idea as to whether the approach is applicable to a specific NP/protein system.

**Shape evaluation of NP-protein complex.** Once obtained, the interaction parameters can be placed in equation (5) and used to determine the average number of proteins ($N_{avg}$) per NP as a function of the concentration of protein ([BSA]). This back calculation of equation (5) allows to calculate the parameter $N_{avg}$ to be determined for each defined experimental BSA concentration.

As stated before, SV-AUC not only measures sedimentation coefficient of a species but also diffusion coefficient. According to the hydrodynamic scaling law[30], $D$ is a function of $s$ and the frictional ratio ($f/f_0$). The $f/f_0$ is defined as the ratio of translational frictional coefficient of a species and that of an equivalent

spherical substance of the same volume. Hence, the higher the $f/f_0$, the more the substance is non-spherical. Using the scaling law, $f/f_0$ can be written as a function of $D$ and $s$ (with $k_B$ and T being Boltzmann constant and temperature, respectively)[21]:

$$\frac{f}{f_0}(s, D) = \left(\frac{\sqrt{2}}{18\pi}\right)^{2/3} \left(\frac{k_B T}{D}\right)^{2/3} \frac{s^{-1/3}}{\eta} (\rho_{cx} - \rho_s)^{1/3} \quad (7)$$

At each BSA concentration, the density of NP–protein complexes, calculated through equations (3) and (5), were used in equation (7) and the corresponding $f/f_0$ values were calculated accordingly. The resulting values were plotted against BSA concentration and $N_{avg}$ as shown in Fig. 4.

Recent studies determined the hydrodynamic shape of BSA as a triangular prism of an equilateral triangle with 8.4 nm edge length and a thickness of 3.2 nm (ref. 31). If we consider the side-on attachment of BSA to AuNPs consistent with the maximum interaction geometry[8], for large particles such as citrate-AuNPs ($d_{core} = 13$ nm), spherical geometry would be largely preserved on binding of BSA because of the small change in overall diameter. Accordingly, by estimating $f/f_0$ for each protein addition to citrate-AuNPs, it was possible to see slight fluctuations in the spherical value of 1.2 but no systematic trend (Fig. 4a). Conversely, when $f/f_0$ values were estimated for medium sized-NPs-BSA complexes (namely MUA-AuNPs), there was a gradual increase on the first additions of BSA up to a $f/f_0$ value of $\sim 1.6$, followed by a decrease reaching a plateau value around 1.2. It is interesting to note that the peak value arises when the $N_{avg}$ is around two proteins (Fig. 4b). This suggests that after the first

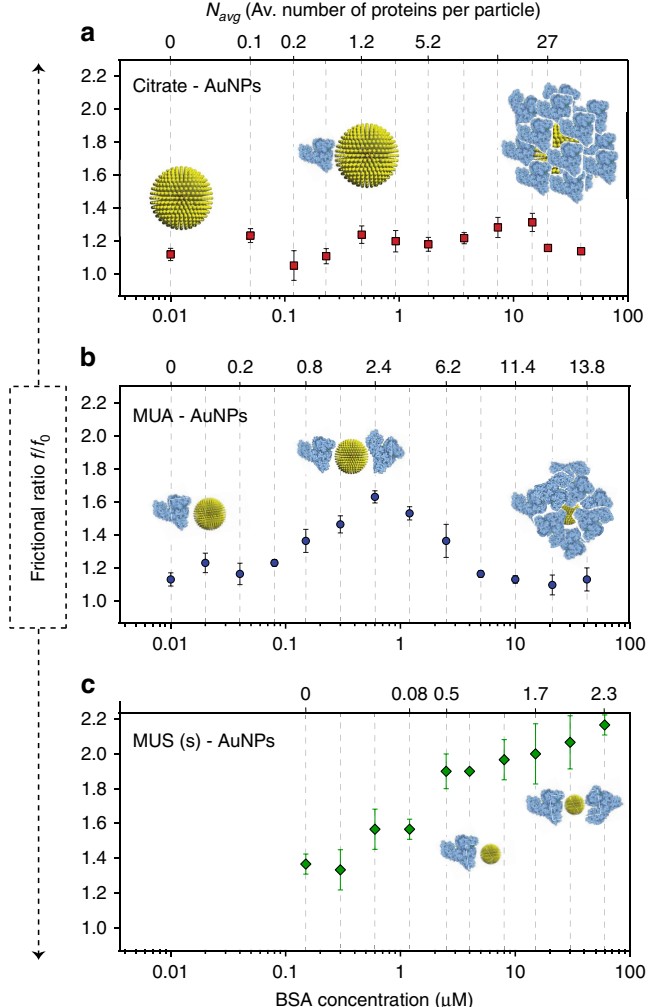

**Figure 4 | Anisotropic shape evolution of NP-protein complexes.** Plots of frictional ratio versus BSA concentration (bottom x-axis) and $N_{avg}$ (top x-axis) for (**a**) Citrate–AuNPs, (**b**) MUA–AuNPs and (**c**) MUS(s)–AuNPs. Possible arrangements of proteins on a nanoparticle are depicted in the plots to clarify the anistropic change on protein binding. It should be noted that these are only cartoon images and not based on scientific simulation. Error bars represent the standard deviation of three independent measurements from the same batch of NPs.

BSA molecule was attached to the NP, the second one approaches from the other side of the complex due to the steric constraints and created an elongated structure. When further BSA molecules were introduced, regardless of the binding position, they gradually reconstructed the complex's globular shape until NPs were fully covered, in which case $f/f_0$ was again 1.2.

In case of very small MUS(s)–AuNPs, the interaction parameters indicate $N_{max}$ to be around 2 BSA molecules per particle at saturation (Table 3). The analysis of $f/f_0$ revealed that there was an increase in $f/f_0$ starting right after the first additions of BSA and this continued until the final addition of BSA. In the case of prolate assembly of two proteins with one particles, one could speculate that this is formed under centrifugal forces. Even though we cannot readily rule out this possibility, we should point out that the effect of the centrifugal forces on proteins' alignment on the NP surface would be visible in the sedimentation data analysis. Any change in $f/f_0$ during the measurement would lead to systematic error in the Lamm equation fitting which was never observed. Moreover, this type of alignment would happen only if

the particle-protein assembly does not freely rotate during the centrifugation. On the contrary, we think the assemble does rotate freely, due to Brownian motion dominating at this length scale[32], as shown by the large diffusion coefficients we observed. In addition, gold nanorods of comparable dimension have shown not to align under centrifugal forces (because of Brownian motion)[33].

Speculations about the mechanism are similar to the ones presented in the MUA–AuNPs' case but with stronger effects on the $f/f_0$ as the size of the particles is more comparable to the size of the proteins (with the saturation at ~2 proteins per particles). Overall, frictional ratio analysis of particularly small particles–protein conjugates suggests that the complex can deviate considerably from its' original sphere. This analysis cannot be considered analytically rigorous as some of the thermodynamic association parameters are first derived based on the assumption of $f/f_0$ to be ~1 and then used to find deviation of $f/f_0$ from 1. Yet, we believe that the overall indication is correct (and strongly backed up by geometrical arguments). A more rigorous approach could be developed with multi-wavelength AUC. Future studies with multi-wavelength AUC could improve the understanding of the NP–protein system by adding wavelength dimension to the analysis. Monitoring protein and NP concentration separately could allow more information on the NP-protein interactions to be obtained[34]. For example, the density of NP-protein complex is solely a theoretical approximation based on individual NP and protein densities, which decreases the accuracy of the $f/f_0$ analysis. Instead, recent advancements in multi-wavelength AUC could provide much more reliable information on the axial ratio of NP-protein complexes through multidimensional analysis. This could also allow us to analyse protein interaction with other types of nanomaterials such as gold nanorods and carbon nanotubes. Finally, the applicability of the method could be widened through the employment of other optical sources in AUC, such as an interferometer. Characterizing the species inside the solution by fringes stemming from refractive index changes offers a convenient workaround for non-absorbing soft materials like polymers and dendrimers.

With the method developed here, it was possible to accurately determine dissociation constants ($K_D$), maximum protein number per NP ($N_{max}$), and Hill coefficient ($n$) for the association of proteins to monodisperse NPs regardless of their size. The strength of the methods is in its label-free nature that allows for the study of large or small particles with no increase in complexity. Additionally, there is minimal influence from free-proteins and from possible protein aggregates in the medium.

## Methods

**NP synthesis.** Citric acid trisodium salt (>98%), 11-mercaptoundecanoic acid (98%), 1-octanethiol (>97%), gold(III) chloride trihydrate (>99.9%), sodium borohydride (99%), chloro(triphenylphosphine) gold(I) and borane tert-butylamine complex (97%) were purchased from Sigma Aldrich and used without further purification. 11-mercaptoundecane sulfonate sodium salt was synthesized in-house according to a previously reported protocol[35]. All solvents were purchased from Sigma Aldrich with ACS grade purity and no further purification was applied before use.

In general, each AuNPs was synthesized through reduction of gold salts in the presence of water soluble capping agents. For citrate–AuNPs synthesis, an aqueous gold chloride solution (1 ml, 25 mM) was injected into an aqueous sodium citrate solution (150 ml, 2.2 mM) at 100 °C. After 15 min of stirring, the solution was brought to room temperature. To prepare the samples for DGU fractionation, the NP solution was directly concentrated using Amicon Ultra MWCO 30 kDa centrifugal filters in Sorvall Legend XT/XF Thermofisher Centrifuge at 6,000 rcf for 25 min.

MUA-AuNPs were synthesized according to a previously reported protocol with the exception that MUA was used instead of MUS/OT capping ligands[36]. MUS(s)–AuNPs synthesis followed previously reported methods[37] and large particles and aggregates were removed via DGU fractionation.

**NP DGU fractionation.** After synthesis of AuNPs, they were subjected to DGU fractionation to remove aggregates and increase the homogeneity of size distribution. Approximately 500 µl of NP solution (10 mg ml⁻¹) was deposited on top of the continuous sucrose gradient (20–50% w/v) which was prepared with Gradient Station (Biocomp Instruments, 38.5 ml Ultraclear tubes). Then, the tubes were placed in SW 28 Ti Swinging Bucket Rotor (Beckman Coulter) and spun at 28 k r.p.m. for at least 1 h. AuNPs that were spread through the tube were collected with 4–10 mm zone intervals by using Piston Operated Fractionator (Biocomp Instruments). All fractions were thoroughly cleaned from sucrose with Amicon Ultra MWCO 3 kDa centrifugal filters with Sorvall Legend XT/XF Thermofisher Centrifuge at 6,000 r.c.f. for 25 min. Refer to Supplementary Fig. 4 for photographic comparison for before/after fractionation of MUS(m)–AuNPs. AUC analysis was carried out for each NPs after fractionation to assess the monodispersity (Supplementary Figs 5 and 6).

**Analytical ultracentrifugation.** Bovine Serum Albumin was purchased from Fisher Scientific (Bioreagent grade, lyophilized powder). All BSA solutions were prepared freshly in 10 mM NaCl or PBS. All NPs were added to BSA solutions such that the final solution had 0.8–1.8 OD absorbance at 520 nm in AUC sapphire cell (path length is 1.2 cm). Before AUC measurements, these solutions were incubated at 4 °C overnight without stirring. To obtain enough scans (40–50, at 20 °C) before complete sedimentation of species, AUC measurements were done under varying speeds from 6,000 to 12,000 r.p.m. with An50 Ti Rotor (Beckman Coulter). Data analysis was conducted with SEDFIT software and weighted average sedimentation and diffusion coefficients were calculated with a custom made MATLAB program that was previously reported[20].

**Data availability.** The raw data of all AUC experiments described here can be found through the following link: https://figshare.com/s/3d859007498efbfda8b2.

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

## Acknowledgements

This work was financially supported by the Self-Assembled Virus-like Vectors for Stem Cell Phenotyping (SAVVY) project within the European 7th Framework Programme and FutureNanoNeeds the project with the European H2020 Framework Programme. We acknowledge support from Paulo Henrique Jacob Silva for MUS ligand synthesis. We also thank Dr Emma-Rose Janecek for her invaluable help on proofreading the manuscript.

## Author contributions

A.B. and F.S. designed the experiments and discussed the results. A.B. carried out all the experiments. The manuscript was written with contributions from both authors. Both authors have given approval to the final version of the manuscript.

## Additional information

**Competing financial interests:** The authors declare no competing financial interests.

