## [Peer Review File · Nature Communications]

Reviewers' comments:

Reviewer #1 (Remarks to the Author):

This manuscript describes the use of analytical ultracentrifugation (AUC) as an absorbance-based tool to quantitatively study nanoparticle-protein interactions. With this approach, they analyzed the interactions between bovine serum albumin and gold nanoparticles with different sizes (core diameter: 2.2-12.6 nm) and surface ligands, where parameters including equilibrium constants, stoichiometry and Hill coefficients were obtained and compared.

The proposed method, AUC, is not new, neither is the studied system or the obtained result. The new aspect of the present work is the use of AUC to quantitatively characterize nanoparticle-protein interactions. The manuscript gives a proof of principle of the method using a single protein, serum albumin.

While the results appear reasonable, the reviewer does not see a clear advantage over other established techniques for model studies of this kind.

Overall, this work is recommended to be published in one of the more specialized nanoparticle research journals after revisions as suggested in the following:

- (1) There is something wrong with numbering of figures (two Fig. 1). Please check.
- (2) The molar concentrations of nanoparticles used in the protein binding experiment are not specified. In the Hill adsorption isotherm equation, the concentration of free proteins is only approximately equal to the totally added proteins when the protein is in large excess to the amount of nanoparticles. Or else, the obtained binding parameters may deviate from true their values (signature: Hill coeff. increases artificially).
- (3) In Table 3, the values of Hill coefficient (n) in the last three types of nanoparticles are clearly larger than 1, which normally suggest a cooperative binding behavior. How to explain this unusual cooperative binding of BSA/HSA protein to nanoparticles?
- (4) In Fig. S4, authors performed a dilution experiment to show that their system is reversible, thus to validate the use of Hill equation. However, the concentrations of protein they used before (10 mM) and after (1 mM) the dilution are all within the saturation region, according to the binding curve shown in Fig. 1B (line 202-203). (Note that the figure S4 does not specify the NP, but the text (para starting with line 186) implies that it should be MUS(m)-AuNPs ($d_H=7.6$ nm).) In this case, one can not obtain any conclusions regarding the reversibility of the system.

Reviewer #2 (Remarks to the Author):

The authors (to my knowledge for the first time) show the sequential addition of single proteins to nanoparticles surfaces as the dispersion parameters are varied. While a number of previous discussions have shown an increase in size with formation of the corona, this is a quite new level of insight into the corona formation process. Possibly the most striking outcome is that one has sufficient resolution to actually 'see' shapes of complexes, and one can foresee many uses for this in future.

From the larger point of view, the ideas, methods and approaches are in my view all correct. I make only the comment that the degree of reversibility (upon which the Hill equation is predicated) will depend also on the systems chosen. Some proteins for larger particles will not share this property, and some of the 'hard corona's' are very firmly fixed and irreversible. The fact that the authors have thought to check this in their system is a mark of care. However, I would mention in the manuscript this caution, for some of those that follow may not consider this aspect, and take that for granted.

Separately, I would also check just to be sure that the actual gravitational forces involved are themselves not sufficient to cause the opposite side alignment for two proteins added to the sphere. I have done a small calculation of this myself and believe it is not the cause, but the authors maybe could be sure of that.

The paper is also well written and clear. In my opinion it could use a few extra small illustrations/sketches that clarify the shapes being observed, but this is a very minor issue. Some of the organization of the equations (they being quite long) is also a little bit complicated and hard to read, but again, a minor issue likely corrected when the paper is printed.

I find the paper to be most interesting, and believe it will have widespread interest. It should be published.

minor comments:

Line 132 Equation (6) is floating the parameters KD , N_{max} : incorrect use of English

Reviewer #3 (Remarks to the Author):

This is an excellent work reporting on the quantitative evaluation of the interaction of bovine serum albumin as a model protein with gold nanoparticles with varying size and surface functionalization. The evaluation of such interactions is of great importance in the field of nanotoxicology and others. Therefore, already earlier studies exist which are based on fluorescence labelling of the proteins or scattering techniques. But both of these approaches have their significant limitations as discussed in the introduction of this work. Using analytical ultracentrifugation (AUC) as a fractionating method, this can now be overcome and for the first time, a label free technique, which is able to selectively fractionate and detect the particles of interest becomes available. Using a Langmuir adsorption isotherm approach, it is possible to fit the BSA concentration dependence of the average sedimentation coefficient to obtain the dissociation constant and the number of interacting BSA molecules. This method works very well as demonstrated for different Au nanoparticles. Also, the reversibility of the BSA adsorption, which is a prerequisite for the evaluation could be shown. I therefore recommend this important work for publication in Nature Communications and am sure that it will be of significant impact if a few minor issues can be solved as noted below.

- p4 introduction: The websites for SEDFIT and UltraScan should be cited when these programs are mentioned.
- Eqs 4 & 5: I believe the cubic root should be drawn until the end of the equation
- Tab. 1: It is unclear how exactly the core diameter and the hydrodynamic diameter were derived. Which density was used to calculate dH ? In addition, dH is not always the sum of core diameter + 2 x ligand length but sometimes even larger and in addition, it seems unlikely that the ligand is fully stretched as calculated. Therefore, this issue needs consideration and discussion.
- There are actually 2 x Fig. 1. Please number the second figure as Fig. 2 and update the rest of the manuscript accordingly.

- Please provide a citation for eq. 7
- It would be much more advantageous if f/f_0 would be related to the corresponding axial ratios for prolate ellipsoids of revolution since then, actual a/b values would be obtained, which would help a lot in the discussion of the actual shapes of the nanoparticles after BSA adsorption. This conversion is for example possible using UltraScan but there are other programs as well.
- It would be beneficial for the reader if the application range of the introduced method could be outlined in terms of particle sizes and range of kD 's
- Fig. SI4. It would be good if dashed lines could be included giving the average sedimentation coefficient for 10 μ M BSA = 65 S according to Fig 2 (not 10 mM as stated in the legend of SI 4 ?) and the same for 10 x diluted = 1 μ M BSA = 75 S. The average sedimentation coefficients of the distributions in the figure should be stated so that a comparison to Fig. 2 is possible to show the reversibility.
- The English sounds a bit odd in some parts and the paper should be proofread

Answers to Reviewers (blue original text from the reviewers, black responses, red new text in the paper):

Reviewer #1:

The proposed method, AUC, is not new, neither is the studied system or the obtained result. The new aspect of the present work is the use of AUC to quantitatively characterize nanoparticle-protein interactions. The manuscript gives a proof of principle of the method using a single protein, serum albumin.

While the results appear reasonable, the reviewer does not see a clear advantage over other established techniques for model studies of this kind.

We agree with the referee that the main part of this paper is the quantitative determination of nanoparticles-protein interactions. There are, in our opinion, a few novel aspects in this work, for example the measurement is absorption-based and hence does not lose accuracy upon protein aggregation. Also because of its nature (absorption and ultracentrifugation) the measurement is ideally suited to quantify these parameters for very small particles, something very hard to do otherwise.

There is something wrong with numbering of figures (two Fig. 1). Please check.

We corrected the numbering accordingly.

The molar concentrations of nanoparticles used in the protein binding experiment are not specified. In the Hill adsorption isotherm equation, the concentration of free proteins is only approximately equal to the totally added proteins when the protein is in large excess to the amount of nanoparticles. Or else, the obtained binding parameters may deviate from their true values (signature: Hill coeff. increases artificially).

We thank the referee for this comment. We think this comment is extremely valuable and important to understand as to whether the system is applicable. We had checked for this previously but only for one type of particles. While thinking at the referee's comment we realized that (1) the molar concentration of the nanoparticles does depend on their size and hence we decided to specify it in the paper (see below) for each particle, and (2) that the nanoparticle concentration (set by the sensitivity of the optical detector) sets a minimum value on the K_d that our technique can measure (see response below to referee #3).

We calculated the molar concentrations of each type of gold nanoparticles shown in this work using a model now explained in supplemental materials. We have marked this concentration in adsorption isotherm plots. It is shown that nanoparticle concentrations are at least 10 times less than protein concentrations at the point where the binding starts, hence we believe that our treatment is valid.

In Table 3, the values of Hill coefficient (n) in the last three types of nanoparticles are clearly larger than 1, which normally suggest a cooperative binding behavior. How to explain this unusual cooperative binding of BSA/HSA protein to nanoparticles?

While this is a good comment on the measured value, we should point out that a discussion on the interpretation of the data is out of the scope of this paper, instead, it is to present a technique. Of course the data have to be correct and the paper needs to assure correctness of the data. For this reason we had compared the data obtained with known literature values. We agree that cooperative Hill coefficient is worth noticing, we have now provided a few literature examples to show that Hill coefficient larger than 1 had already been measured in nanoparticles-protein interaction and cooperative binding had been postulated.

“...Additionally, Table 3 summarizes Hill coefficients varying depending on the particle type. Citrate and MUA coated AuNPs show anti-cooperative (<1) effect while sulfonate functionalized particles show cooperative effect. A discussion on the meaning of these observations is beyond the scope of this paper, we have to point out that cooperative Hill coefficient have already been observed in some nanomaterials interaction with proteins...^{1,2}”

In Fig. S4, authors performed a dilution experiment to show that their system is reversible, thus to validate the use of Hill equation. However, the concentrations of protein they used before (10 mM) and after (1 mM) the dilution are all within the saturation region, according to the binding curve shown in Fig. 1B (line 202-203). (Note that the figure S4 does not specify the NP, but the text (para starting with line 186) implies that it should be MUS(m)-AuNPs ($d_H=7.6$ nm).) In this case, one can not obtain any conclusions regarding the reversibility of the system.

We apologize for the confusion in this part. First, the NPs used here are now clearly identified in the figure caption.

“...MUS(m)-AuNPs ($d_H = 7.6$ nm) is mixed with 1 μ M BSA solution and incubated 16 hours at 20 $^{\circ}$ C...”

Second, the starting protein concentration was typed wrong, it is actually 1 micromolar. With the correct version, the shift is visible in terms of the sedimentation coefficient of nanoparticle-protein mixture before and after dilution. Additionally, we put the average sedimentation coefficient values before and after dilution into the legend of the plot in SI4 for the reader to identify which region it is in adsorption isotherm.

Reviewer #2:

The authors (to my knowledge for the first time) show the sequential addition of single

proteins to nanoparticles surfaces as the dispersion parameters are varied. While a number of previous discussions have shown an increase in size with formation of the corona, this is a quite new level of insight into the corona formation process. Possibly the most striking outcome is that one has sufficient resolution to actually 'see' shapes of complexes, and one can foresee many uses for this in future.

We thank the referee for this positive assessment of our paper.

From the larger point of view, the ideas, methods and approaches are in my view all correct. I make only the comment that the degree of reversibility (upon which the Hill equation is predicated) will depend also on the systems chosen. Some proteins for larger particles will not share this property, and some of the 'hard corona's' are very firmly fixed and irreversible. The fact that the authors have thought to check this in their system is a mark of care. However, I would mention in the manuscript this caution, for some of those that follow may not consider this aspect, and take that for granted.

We acknowledge this point and agree that the reader should not be misguided. We put the necessary comments and relevant references in the manuscript to clarify this issue further.

“It is indispensable, on the other hand, to point out that not every type of protein – nanoparticle interactions will show this (or any) degree of reversibility. It is reported that some larger polystyrene particles show irreversible hard corona formation with transferrin.³”

Separately, I would also check just to be sure that the actual gravitational forces involved are themselves not sufficient to cause the opposite side alignment for two proteins added to the sphere. I have done a small calculation of this myself and believe is is not the cause, but the authors maybe could be sure of that.

We also agree that it is unlikely for a centrifugal force to affect the positioning of proteins on the surface of nanoparticles especially after binding.

We have performed the following calculations:

The energy needed to move a protein should be

$$E = F * \Delta x = g * m_p * rcf * \Delta x = 9.8 \frac{m}{s^2} * 66.4 \frac{kg}{mol} * 18144 rcf * (\pi * 3 * 10^{-9} m) \\ = 0.035 \text{ Joule/mole}$$

where F is the centrifugal force Δx is the displacement on the nanoparticle (assumed to have a 3 nm radius), and g is the gravitational constant, and rcf is the rotational centrifugal force, and m_p is the mass of the protein.

The approximate binding energy is

$$E = R * T * \ln\left(\frac{K_D}{C}\right) = 8.31 \frac{J}{K.mol} * 293 K * \ln(10^{-6}) = -28 \text{ kJ/mol}$$

where R is the gas constant, T is the temperature, K_D is the dissociation constant, and C is the reference state.

Based on the above result we do not think that there is a possibility for the centrifugal forces to (1) align and (2) move the proteins on the nanoparticles. The calculations do have many profound exemplifications in them; hence we prefer not to speculate in the paper about them. We have added the following sentence to the paper.

“In this case of the prolate assembly of two proteins with one particles, one could speculate that this is formed under centrifugal forces. Even though we cannot simply rule out this possibility, we should point out that the effect of the centrifugal forces upon proteins’ alignment on the nanoparticle surface would be visible in the sedimentation data analysis. Any change in f/f_0 during the measurement would infer systematic error in the Lamm equation fitting which is never observed. Second, this type of alignment would happen only if the particle-protein assembly do not freely rotate during the centrifugation. On the contrary, we think this is the case, due to Brownian motion that dominates at this length scale,³² as shown by the large diffusion we observed. Additionally, gold nanorods of comparable dimension have shown not to align under centrifugal forces (because of Brownian motion).³³”

The paper is also well written and clear. In my opinion it could use a few extra small illustrations/sketches that clarify the shapes being observed, but this is a very minor issue. Some of the organization of the equations (they being quite long) is also a little bit complicated and hard to read, but again, a minor issue likely corrected when the paper is printed.

We modified a few things in the manuscript including an addition of illustrations of nanoparticle-protein conjugates in the f/f_0 vs. [BSA] plot.

“...The possible arrangements are depicted in the plots to clarify the anisotropic change upon protein binding. It should be noted that these are only cartoon images and not based on scientific simulation...”

I find the paper to be most interesting, and believe it will have widespread interest. It should be published.

Line 132 Equation (6) is floating the parameters K_D , N_{max} : incorrect use of English

We changed this sentence.

“In Equation 6 the parameters K_D , N_{max} and n are varied while all the other parameters...”

Reviewer #3:

This is an excellent work reporting on the quantitative evaluation of the interaction of bovine serum albumin as a model protein with gold nanoparticles with varying size and surface functionalization. The evaluation of such interactions is of great importance in the field of nanotoxicology and others. Therefore, already earlier studies exist which are based on fluorescence labelling of the proteins or scattering techniques. But both of these approaches have their significant limitations as discussed in the introduction of this work. Using analytical ultracentrifugation (AUC) as a fractionating method, this can now be overcome and for the first time, a label free technique, which is able to selectively fractionate and detect the particles of interest becomes available. Using a Langmuir adsorption isotherm approach, it is possible to fit the BSA concentration dependence of the average sedimentation coefficient to obtain the dissociation constant and the number of interacting BSA molecules.

We thank the referee for these encouraging words.

p4 introduction: The websites for SEDFIT and UltraScan should be cited when these programs are mentioned.

We added both websites as references.

- “23. Demeler, B. UltraScan. <http://ultrascan.uthscsa.edu> (2015).
24. Schuck, P. P. SEDFIT. <http://www.analyticalultracentrifugation.com> (2016).”

Eqs 4 & 5: I believe the cubic root should be drawn until the end of the equation

We have checked the equations one more time and we think that they are correct as written, that is the cubic root does not extend all the way to the end. The referee is correct that the whole argument should be in cubic root but we had already taken it out by performing the necessary operations.

Tab. 1: It is unclear how exactly the core diameter and the hydrodynamic diameter were derived. Which density was used to calculate d_H ? In addition, d_H is not always the sum of core diameter + 2 x ligand length but sometimes even larger and in addition, it seems unlikely that the ligand is fully stretched as calculated. Therefore, this issue needs consideration and discussion.

For calculation of hydrodynamic diameter and density of nanoparticles, we followed the method described by Carney *et.al.* Nat. Commun. (2011). Basically, Stokes-Einstein equation is applied here. This gave us the hydrodynamic diameter including the hydration shell. We totally agree that simple addition of extended ligand length to core diameter does not represent the real hydrodynamic information. Those ligand lengths are provided only for comparison. Core diameters are obtained by combination of TEM

images and calculation through AUC derived density and hydrodynamic diameters of nanoparticles with an approximate ligand density. With the help of referee's suggestions, we also indicated this in the caption of the table where we present these values.

"...Hydrodynamic diameters and densities of AuNPs are calculated according to previously reported method.⁴ Ligand length information is only provided for comparison. They are not used in any of the calculation in this work."

There are actually 2 x Fig. 1. Please number the second figure as Fig. 2 and update the rest of the manuscript accordingly.

We corrected this.

Please provide a citation for eq. 7

The following reference is added:

"21. Brown, P. H., Balbo, A. & Schuck, P. Characterizing Protein-Protein Interactions by Sedimentation Velocity Analytical Ultracentrifugation. onlinelibrary.wiley.com (John Wiley & Sons, Inc., 2001). doi:10.1002/0471142735.im1815s81"

It would be much more advantageous if f/f_0 would be related to the corresponding axial ratios for prolate ellipsoids of revolution since then, actual a/b values would be obtained, which would help a lot in the discussion of the actual shapes of the nanoparticles after BSA adsorption. This conversion is for example possible using UltraScan but there are other programs as well.

At present, given that we have not been able to find any other technique to confirm our interpretation of f/f_0 , we would prefer not to speculate further on this observation.

It would be beneficial for the reader if the application range of the introduced method could be outlined in terms of particle sizes and range of K_D 's.

We agree with this request of the referee. Limitations and application range is discussed further in the manuscript.

"Overall, this approach could be used for most types of nanoparticles as long as they are suitable for AUC optics systems – absorbance, fluorescence, interference, etc. It is hard to provide exact ranges of binding constants and nanoparticles sizes where this technique can be usable, as it basically depends on the difference in density between particles and proteins. Any size range, then, should be specified with respect to the core material of the nanoparticle. As an indication for gold nanoparticles, we can state that the lower limit of a K_D measurable would be $\sim 0.01 \mu\text{M}$. This is because at this value the

binding of the proteins will happen at approximately the nanoparticles concentration, invalidating one key assumption in Hill equation that the concentration of the proteins is in large excess. Given that the nanoparticle concentration is determined by their absorbance (the optical density of the solution should be in the range of 0.2 to 1.2 for a reliable optical measurement), as particle concentration varies, so does the lower limit for K_D . The upper limit of K_D is mostly independent of nanoparticles but depends on the nature of the protein. One should keep the viscosity of the protein solution in mind, as viscosity could substantially affect the process of sedimentation of the solutes. Based on this consideration we arbitrarily estimate an upper solution concentration of 10 mM. We estimate this upper limit K_D value to be around 1 mM. As discussed above, the limitation of the method in terms of size of nanoparticles is directly related to colloidal stability of the nanoparticles. In our experience gold nanoparticles from 2 to 25 nm core size are easily measurable in AUC. We believe the above discussion can give the reader an idea as to whether the approach is applicable to a specific nanoparticle/protein system.”

Fig. SI4. I would be good if dashed lines could be included giving the average sedimentation coefficient for 10 uM BSA = 65 S according to Fig 2 (not 10 mM as stated in the legend of SI 4 ?) and the same for 10 x diluted = 1 uM BSA = 75 S. The average sedimentation coefficients of the distributions in the figure should be stated so that a comparison to Fig. 2 is possible to show the reversibility.

(See reply to referee #1) We corrected the caption of SI4. There was a typo in the protein concentration we provided: the true concentration is 1 micromolar BSA. We added the average sedimentation coefficient values of both before and after dilution to the legend of SI4. These values are in good match with Fig. 2 values.

“...MUS(m)-AuNPs ($d_H = 7.6$ nm) is mixed with 1 μ M BSA solution and incubated 16 hours at 20 °C...”

The English sounds a bit odd in some parts and the paper should be proofread

We have proofread the paper.

REVIEWERS' COMMENTS:

Reviewer #2 (Remarks to the Author):

I believe the authors have found a better way of addressing the issues on role of gravity, local shear forces and movements within the gravitational field than only the simple calculations I had pointed towards. Bearing in mind carefully again that the process is reversible on a characteristic time-scale (that is clear from the data, though the original paper contained a typo in concentration) this point about orientation is not entirely trivial. I think the choice of wording they suggest is prudent for the current paper.

The paper has been improved by response to all of the reviewers, and I think it is a key and novel step in understanding the particle-surface adsorption processes. I would publish it.

Reviewer #3 (Remarks to the Author):

I have looked at the revised paper. The authors have carefully addressed and solved all referee comments. This is true for my comments as well as those of referee 1 & 2. The manuscript has gained quality by these revisions and I recommend that the paper shall be accepted now. The authors should nevertheless check the changes, which they made for typos and English.